# Multicenter Technical Validation of 30 Rapid Antigen Tests for the Detection of SARS-CoV-2 (VALIDATE)

**DOI:** 10.3390/microorganisms9122589

**Published:** 2021-12-15

**Authors:** Gilbert Greub, Giorgia Caruana, Michael Schweitzer, Mauro Imperiali, Veronika Muigg, Martin Risch, Antony Croxatto, Onya Opota, Stefanie Heller, Diana Albertos Torres, Marie-Lise Tritten, Karoline Leuzinger, Hans H. Hirsch, Reto Lienhard, Adrian Egli

**Affiliations:** 1Institute of Microbiology, Lausanne University Hospital, University of Lausanne, 1011 Lausanne, Switzerland; Giorgia.Caruana@chuv.ch (G.C.); Antony.Croxatto@ne.ch (A.C.); Onya.Opota@chuv.ch (O.O.); 2Infectious Diseases Service, Department of Internal Medicine, Lausanne University Hospital, University of Lausanne, 1011 Lausanne, Switzerland; 3Coordination Commission of Clinical Microbiology, Swiss Society of Microbiology, 1033 Cheseaux, Switzerland; martin.risch@risch.ch (M.R.); Reto.Lienhard@ne.ch (R.L.); Adrian.Egli@usb.ch (A.E.); 4Clinical Bacteriology and Mycology, University Hospital Basel, 4031 Basel, Switzerland; michael.schweitzer@usb.ch (M.S.); veronika.muigg@usb.ch (V.M.); stefanie.heller@unibas.ch (S.H.); diana.albertostorres@unibas.ch (D.A.T.); 5Applied Microbiology Research, Department of Biomedicine, University of Basel, 4031 Basel, Switzerland; 6Centro Medicina di Laboratorio Dr Risch, Via Arbostra 2, 6963 Pregassona, Switzerland; mauro.imperiali@risch.ch; 7ADMed Microbiologie Laboratory, 2300 La Chaux-de-Fonds, Switzerland; Marie-Lise.Tritten@ne.ch; 8Clinical Virology, University Hospital Basel, 4031 Basel, Switzerland; Karoline.Leuzinger@usb.ch (K.L.); hans.hirsch@unibas.ch (H.H.H.); 9Transplantation & Clinical Virology, Department of Biomedicine, University of Basel, 4031 Basel, Switzerland; 10Infectious Diseases and Hospital Epidemiology, University Hospital Basel, 4031 Basel, Switzerland

**Keywords:** SARS-CoV-2, COVID-19, rapid antigen test, diagnostics, virus testing

## Abstract

During COVID19 pandemic, SARS-CoV-2 rapid antigen tests (RATs) were marketed with minimal or no performance data. We aimed at closing this gap by determining technical sensitivities and specificities of 30 RATs prior to market release. We developed a standardized technical validation protocol and assessed 30 RATs across four diagnostic laboratories. RATs were tested in parallel using the Standard Q^®^ (SD Biosensor/Roche) assay as internal reference. We used left-over universal transport/optimum media from nasopharyngeal swabs of 200 SARS-CoV-2 PCR-negative and 100 PCR-positive tested patients. Transport media was mixed with assay buffer and applied to RATs according to manufacturer instructions. Sensitivities were determined according to viral loads. Specificity of at least 99% and sensitivity of 95%, 90%, and 80% had to be reached for 10^7^, 10^6^, 10^5^ virus copies/mL, respectively. Sensitivities ranged from 43.5% to 98.6%, 62.3% to 100%, and 66.7% to 100% at 10^5^, 10^6^, 10^7^ copies/mL, respectively. Automated assay readers such as ExDia or LumiraDx showed higher performances. Specificities ranged from 88.8% to 100%. Only 15 of 30 (50%) RATs passed our technical validation. Due to the high failure rate of 50%, mainly caused by lack of sensitivity, we recommend a thorough validation of RATs prior to market release.

## 1. Introduction

The SARS-CoV-2 pandemic has lead to an unprecedented burden of individual and public health [1]. SARS-CoV-2 diagnostic assays became the corner stone of patient care and epidemiological management. Diagnostic laboratories reacted promptly, adapting workflows according to demands [2]. However, in many countries high case numbers resulted in limited reagents and focus on severely ill patients. In this situation, alternative testing strategies using SARS-CoV-2 specific rapid antigen testing (RATs) presented a promising opportunity for individual diagnostics and population screenings. RATs mostly target abundant viral proteins such as the SARS-CoV-2 nucleocapsid (N)-protein, and rarely the spike (S)-protein [3]. In general, RATs are cheaper, easy to perform, and faster than most molecular methods. However, analytical sensitivities are lower compared to RT-PCRs [4]. Moreover, RATs with a lower specificity than RT-PCR have been used increasingly in clinical settings with a low pre-test probability. This resulted in a low positive predictive value [5] demanding subsequent RT-PCR confirmation.

For several reasons a thorough evaluation of SARS-CoV-2 specific RATs is critical. A plethora of RATs were rapidly released on the market. The viral pathogen was still relatively new and many production companies and their product quality were unknown. Furthermore, the poor performance of RATs for other respiratory viruses such as Influenza has been known for decades [6,7,8]. Finally, diagnostic assay performance including sensitivity and specificity depends on many pre-analytical factors, such as the assay user, sample material and clinical settings, e.g., symptomatic or asymptomatic patients and time since symptoms onset [4,8,9,10,11,12,13,14,15,16,17,18,19,20].

The Federal Office of Public Health (FOPH) in Switzerland authorized the introduction of RATs in November 2020 based on clinical and technical validation criteria and on the recommendations of the Swiss Society of Microbiology [20]. Thus, the FOPH mandated the Swiss Society of Microbiology to determine the sensitivity and specificity of RATs prior to market release using a shared validation protocol. We present the results of the national multicenter validation of 30 SARS-CoV-2 RATs performed across four diagnostic laboratories.

## 2. Materials and Methods

### 2.1. Study Design

We developed a technical validation protocol as part of the Coordination Commission for Clinical Microbiology of the Swiss Society for Microbiology (File S1 in [20]) and performed a technical validation with every center using the same standardized protocol for 30 RATs. Appendix A lists the used assays.

### 2.2. Samples

We used left-over material collected with flocked swabs from nasopharyngeal swabs and sent to the laboratory for SARS-CoV-2 specific RT-PCR testing. Samples were suspended in 1–3 mL of transport media. For each RAT, we collected 100 SARS-CoV-2 RT-PCR positive samples and 200 SARS-CoV-2 RT-PCR negative samples for the validation. All samples were stored at 4 °C and used within 72 h from collection. The only exceptions were reference samples shared between all centers (Appendix A). Five highly positive reference samples were included. We diluted samples in transport media, and shipped frozen aliquots to each center. Similarly, 50 negative reference samples were included. These originated from a respiratory virus biobank collected at the University Hospital Basel between January 2017 to October 2020 and were stored at −80 °C until usage in this validation. The collection contained diverse respiratory viruses to assess cross reactivity of RATs, including: four human coronaviruses (HCoV-229E, HCoV-HKU1, HCoV-OC43, and HCoV-NL63), parainfluenza viruses 1 to 4, rhino/enterovirus, influenza A and B, respiratory syncytial virus (RSV), and human metapneumovirus. The samples were pooled, aliquoted, and shipped frozen to each center. All reference samples were evaluated with the Biofire^®^ FilmArray^®^ Respiratory panel (bioMérieux, Marcy-l’Etoile, France), a SARS-CoV-2 specific quantitative RT-PCR and a virus specific quantitative RT-PCR.

### 2.3. Validation Protocol

Details of the protocol are described in File S1 in [20]. First, a technical validation of at least 300 samples was conducted. Then, we determined the limit of detection from two positive samples using a serial dilution (see “Limit of detection”). Of note, the second step was not considered in the evaluation of the performance for the FOPH. The different assays were batch-wise evaluated, usually up to four different RATs at the same time. In each batch, the Standard Q^®^ (SD Biosensor, Suwonsi, Republic of Korea/ Hoffmann, Roche, Basel, Switzerland) was used as internal reference standard. The Standard Q^®^ (as well as the Panbio^TM^ COVID-19 Ag Rapid Test from Abbott, Lake Forest, IL, USA) was validated in previous clinical studies [10,18] and already commercialized at the time of the beginning of the present study. In addition, results from a technical validation of the Panbio^TM^ assay were presented elsewhere [4]. The Standard Q^®^ was chosen as internal reference based on its more frequent use.

### 2.4. Technical Validation

Transport media and respective buffer solution from the RAT were mixed in a 1:1 ratio. From this mixture the assay was performed according to the manufacturer’s instructions. Three to four drops were added to the flow device and reading was done after 10 to 20 min. Reading was performed visually or with specific reading devices, according to the instrument protocol.

### 2.5. Limits of Detection

To determine the limit of detection, two highly positive samples were diluted in a 1/2 dilution series over 7 steps and each step was tested with the RAT (File S1 in [20]). One of the positive samples was from an infected Vero cell line and the second sample was from a highly positive patient left-over material. These were aliquoted at respective dilution steps, frozen and shipped to each center. For both assessments, detection of a cycle threshold (Ct) between 21.1 (around 2.3 × 10^7^ copies/mL) and 23.3 (around 6.8 × 10^6^ copies/mL) were considered the lower limits of detection.

### 2.6. Reference Method

Confirmation by SARS-CoV-2 specific RT-PCR. RT-PCR confirmation was performed on every sample. Different RT-PCR systems were used as the reference standard method among all laboratories. The most commonly used assay was the cobas6800^®^ SARS-CoV-2 System (Roche, Basel, Switzerland) [21]. Other RT-PCR systems used included: VIASURE SARS-CoV-2 (N1 + N2) Real Time PCR Detection Kit for BD MAX™ (Becton Dickinson, Franklin Lake, NJ, USA); cobas^®^ Liat^®^ System (Roche, Basel, Switzerland); Allplex™ 2019-nCoV Assay (Seegene, Seoul, South Korea); GeneXpert© SARS-CoV-2 test (Cepheid, Sunnyvale, CA, USA); and an in-house platform using Magnapure RNA-extraction coupled to Applied Biosystems 7900 amplification device (QuantStudio 7) [22,23]. Transport media volumes slightly differed: 400 µL of transport media for cobas6800^®^ analyses, 500 µL for VIASURE BD MAXTM, 200 µL for cobas^®^ Liat^®^ and Magnapure platform, and 300 µL for GeneXpert© analyses.

### 2.7. Performance Acceptance Criteria

Initially, we used non-inferiority criteria to the previously validated Standard Q^®^ Rapid Antigen Test test: ≤5% differences in sensitivity and ≤ 0.5% differences in specificity. This was used for the following assays: AMP SARS-CoV-2 Ag (AMP Diagnostics, Graz, Austria), COVID-19 Rapid Antigen Test Cassette (CLUNGENE, Hangzhou, China), Exdia COVID-19 Ag (Precision Biosensor, Inc., Daejeon, Republic of Korea), The BD Veritor™ System (Becton Dickinson, Franklin Lakes, NJ, USA), Lansionbio^®^ COVID-19 Antigen Test Kit—Dry Fluorescence Immunoassay (Lansion Biotechnology Co., Jiangsu, China), Genedia W COVID-19 Ag (Green Cross Medical Science Corp., Giheung-gu, Yongin-si Gyeonggi-do, Republic of Korea) and for COVID-19 Antigen Rapid Test Cassette (BIOZEK, Apeldoorn, The Netherlands). After this, we defined the following acceptance criteria for the technical validation: (i) For samples with Ct values ≤29, ≤26, and ≤23 (approximately 10^5^, 10^6^, and 10^7^ virus copies per mL, respectively) the cumulative sensitivity had to be ≥80%, ≥90%, and ≥95%, respectively. Specificity had to be ≥99%. These criteria resulted in national recommendations from the Swiss Society of Microbiology and published by the FOPH as part of the COVID19 ordinance 3, annex 5a [20,24].

### 2.8. Statistical Analysis

Sensitivity and specificity of each RAT compared to results from SARS-CoV-2 specific RT-PCR were assessed, including overall accuracy and 95% confidence intervals (CI). Sensitivity was stratified by viral loads. We used Kruskal-Wallis H test and multiple pairwise test to compare median Ct among different laboratories. Data were analyzed on “R statistical software” (version 3.6.1, 2019, Vienna, Austria).

### 2.9. Ethical Declaration

This project was prepared according to STANDARD guidelines for diagnostic accuracy studies reporting. The performance data of the different antigen assays were obtained during a quality enhancement project. According to the Swiss Human Research Act, publication of anonymized results of such a quality related project do not require approval of an ethics committee.

## 3. Results

### 3.1. Sample Characteristics

We used 4523 left-over samples to validate 30 RATs between November 2020 and February 2021 across four laboratories. With each sample, we evaluated up to four RATs in parallel and generated 14,544 RAT results. Excluding the internal reference RAT (Standard Q^®^) and invalid results, we obtained 10,021 RAT results. Of note, we further excluded the Clinitest (Siemens, Erlangen, Germany) due to known cross reactivity with the transport media as a technical assessment was not possible with our setup. Hence, out of 9721 valid RAT results, 1804 (18.5%) were RAT positives and 7917 (81.4%) were RAT negatives. In total, 1327 of 4523 (29.3%) samples were SARS-CoV-2 RT-PCR positive. The overall median Ct value was 24.5 (CI: 23.8–25.2; range 9.3 to 38.5), with clearly higher Ct values for negative versus positive RATs, as expected (Figure 1).

Sample distribution across the range of Ct values used was as follows: 563 (42.4%) samples showed a Ct values ≤ 23; 191 (14.4%) samples between > 23 and ≤ 26; 180 (13.6%) samples between > 26 and ≤ 29; and 393 (29.6%) samples > 29. The median Ct values across all four laboratories were 23.2, 27.9, 23.9, and 26.0 for the University Hospital Basel, University Hospital Lausanne, AMED Microbiology, and Dr. Risch Laboratories, respectively (Appendix A).

### 3.2. Sensitivity of RATs

Fifteen out of 30 RATs Passed the Validation (Table 1).

The overall median cumulative sensitivity was 93.5% (95% CI 91.9; 94.7), 87.6% (95% CI 85.9; 89.1) and 76.4% (95% CI 78.6; 81.7) for the Ct ≤ 23 (around 10^7^ copies/mL), Ct ≤ 26 (around 10^6^ copies/mL), and Ct ≤ 29 (around 10^5^ copies/mL), respectively. Figure 2 and Figure 3 show the sensitivity rates for selected RATs. Twelve assays passed the FOPH validation criteria.

Three RATs (AMP Diagnostics, LUNGENE and Becton Dickinson) were validated based on non-inferiority criteria (sensitivity ≤ 5%) to the internal reference. In the end, 15/30 RATs did not pass the validation criteria (Table 2), most of them due to insufficient sensitivity. For Ct values of 29 or higher (roughly 10^5^ copies/mL and lower), RATs generally showed sensitivities below 65% (Table 1 and Table 2, Figure 3A,B).

In contrast, the “top performing” RATs showed a sensitivity above or equal to 90% at this viral load (Table 1, Figure 3C). Noteworthy, the tests validated at Lausanne University Hospital exhibited worse performances likely due to the high proportion of samples taken after four days from symptoms onset among patients tested SARS-CoV-2 positive at the emergency room (data not shown).

### 3.3. Specificity of RATs

The overall median specificity was 98.6%. In between centers, no difference of specificity median values were noted. The overall specificities were 99.9% (range 99.3% to 100%), 96.4% (range 88.8% to 100%), 99.4% (range 99% to 100%), and 99.0% (one test evaluated) the University Hospital Basel, University Hospital Lausanne, ADMED Microbiology, and Dr. Risch Laboratories, respectively. 27/30 (90%) passed the specificity criteria. We observed no cross-reactivity when evaluating the 50 negative samples containing multiple other respiratory viruses (Appendix A).

Internal reference reproducibility. The internal reference RAT was used 13 times on 4523 samples. We evaluated the intra-laboratory and inter-laboratory variability. As shown in Table 3 and Figure 3D, this antigen assay exhibited a specificity ranging from 99.0% to 100%, with a median specificity of 99.1%. The median sensitivity among patients with Ct < 23 was 98.9% (95% CI 98.7; 99.7), ranging from 95.7% to 100%. The median sensitivity among patients with Ct < 26 was 95.1% (95% CI 96.1; 98.0), ranging from 89.5% to 100%. Finally, the median sensitivity among patients with Ct < 29 was 84.3% (95% CI 88.5; 91.6), ranging from 69.4% to 93.7%.

Limits of detection. Analytical sensitivity was tested on two serial dilutions. Minimal lower sensitivity limits were detectable for most assays. Seven RATs (that failed the technical validation) for cell culture supernatant and five RATs (not passing the technical validation) for diluted clinical sample obtained not acceptable results (Appendix A). All tests were also compared to the internal reference RAT, which was always positive above the predefined limit of detection.

## 4. Discussion

Utilizing a shared validation protocol and pre-defined criteria [20,24] allowed us to evaluate a number of RATs within three months. Our work highlights the heterogeneous performances of RATs currently available. Half of 30 investigated assays exhibited an insufficient low sensitivity and/or specificity and did not pass the technical validation. A recent publication also evaluating 29 RATs made a similar observation with strong heterogeneity of performance [25]. The Paul Ehrlich Institute evaluated 122 assays in parallel and also noted a high heterogeneity [26]. Among the 30 tests we evaluated in Switzerland, 9 RATs were considered as “passed” by the Paul Ehrlich Institute in Germany, but failed the validation based on the Swiss criteria reported here and approved by the FOPH [20,24]. Conversely, no test passed the Swiss validation and failed in the Paul Ehrlich validation. Noteworthy, most RATs that clearly failed our technical validation in Switzerland were already used in other countries. Several other validations showed a limited sample size and/or had a selection bias for high viral loads, not reflecting real-world sample distributions. Therefore, many RATs match the WHO thresholds of 80% for sensitivity and 97% for specificity [2]. In addition, only very few studies included other respiratory viruses to evaluate cross reactivity. Our data showed that the false positive rate of 0.5% was not related to cross-reactivity with other respiratory viruses.

The sensitivity of any SARS-CoV-2 assay is dependent on the viral load in the sample [20]. For this reason, we have determined three sensitivity thresholds for Ct values and viral loads. Nevertheless, we noted some differences between centers. First, the investigated population in all laboratories were different, especially in term of the time since symptoms onset. Second, different SARS-CoV-2 PCR systems were used, which could add to difference in Ct values. The internal reference control and the 5% non-inferiority criteria supported the multicenter validation approach by controlling inter-laboratory variability.

This technical validation was done using left-over materials from nasopharyngeal swabs, which are considered the gold standard sample type. Thus, the performance observed here may only be considered for such samples. A reduced sensitivity is likely to be obtained when other sample types such as nasal or buccal swabs or saliva are used [27], but also among asymptomatic subjects [28]. Thus, in summary, the present validation provides a list of tests that may be used on nasopharyngeal samples taken from symptomatic subjects with 1 to 4 days of symptom.

Our study has several limitations. First, we could not control the percentage of COVID-19 asymptomatic patients as we did not access patient charts during this technical validation; second, not every patient included in the study had a COVID-19 symptoms duration shorter than four days; third, one center used multiple RT-PCR methods as reference without transposing the Ct values to those obtained with Cobas; and fourth, we did not define a Ct distribution from the 100 positive samples.

The “freshness” and storage condition of clinical sample may also influence the sensitivity. The viral load determined by RT-PCR may decrease significantly based on storage temperature [29], as we also observed a decrease upon time, while the antigen remained stable to degradation (data not shown). Thus, using non-fresh samples may lead to consider an antigen test more sensitive than its PCR based sensitivity. The problem of faster RNA degradation compared to DNA degradation may be one of the reasons why some manufacturers provide excellent validation reports for tests that clearly exhibit poor performances.

Finally, it needs to be taken into account that visual reading, compared to automated one, could sometimes result subjective, according to the reader’s experience.

## 5. Conclusions

In summary, new RATs should be properly evaluated with (i) a standardized protocol, (ii) sufficient large sample, (iii) declaration on the sample origin, describing the sample material type, symptomatic or asymptomatic patients, and the time since symptoms onset. Overall, the poor sensitivity of multiple tests highlights the general problems with RATs—in critical situations SARS-CoV-2 specific RT-PCR should be used.

## Figures and Tables

**Figure 1 microorganisms-09-02589-f001:**
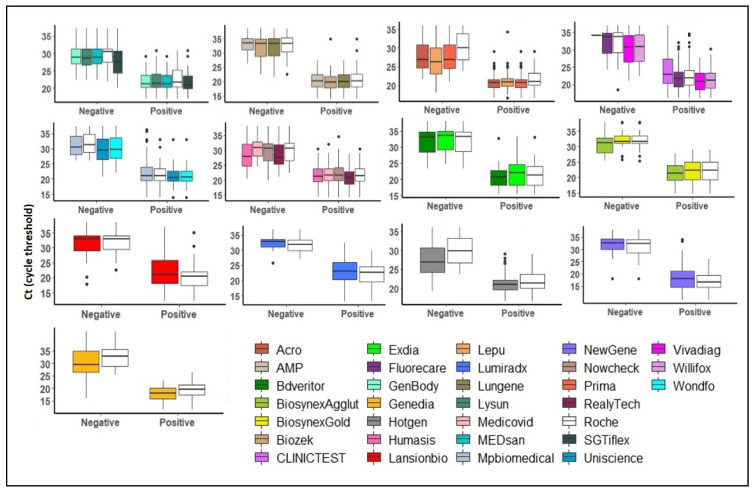
Box-plots of Ct values distributions according to different batches of validation series. Figure 1 legend. RATs were grouped according to the validation series. For each series, there was one internal reference (Standard Q^®^, white box-plot). Black solid lines represent the Ct medians, box-plots and whiskers represent the Ct values distribution. RATs results are displayed on the *x*-axis: Negative or Positive. Cycle thresholds (Ct) are shown on the *y*-axis.

**Figure 2 microorganisms-09-02589-f002:**
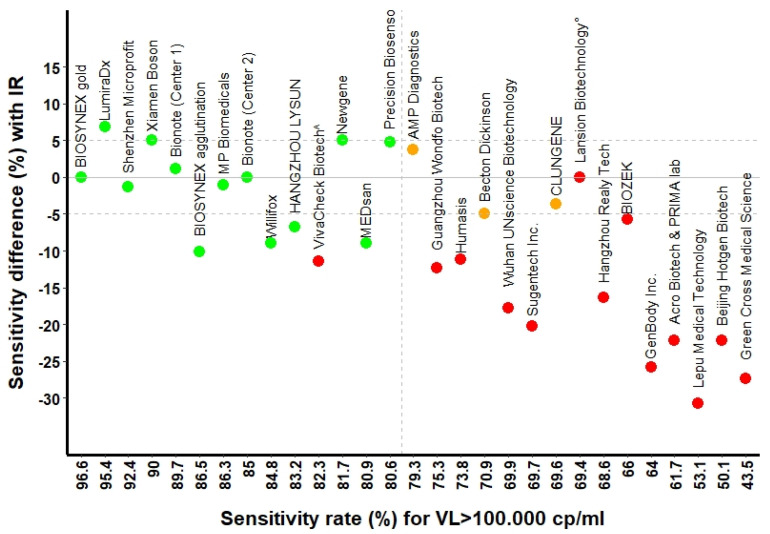
RATs sensitivity rates for viral loads above 105 copies/mL compared to the internal reference test. Figure 2 legend. Green and red dots represent the assays passing and not passing FOPH validation criteria, respectively; orange dots represent the assays validated using the non-inferiority criteria to the internal reference. The solid line represents the internal reference (Standard Q^®^, SD Biosensor/Roche). Dashed horizontal lines represent the limits of difference in percentage within which sensitivity rates’ variations were considered acceptable compared to the IR. The vertical dashed line coincides with 80% cut-off, which was considered the minimal sensitivity threshold for FOPH validation above 105 copies/mL of viral load. ^: this test did not pass the validation criteria for insufficient sensitivity at viral loads above 106–107 copies/mL. °: this test did not pass the validation because of lack of specificity.

**Figure 3 microorganisms-09-02589-f003:**
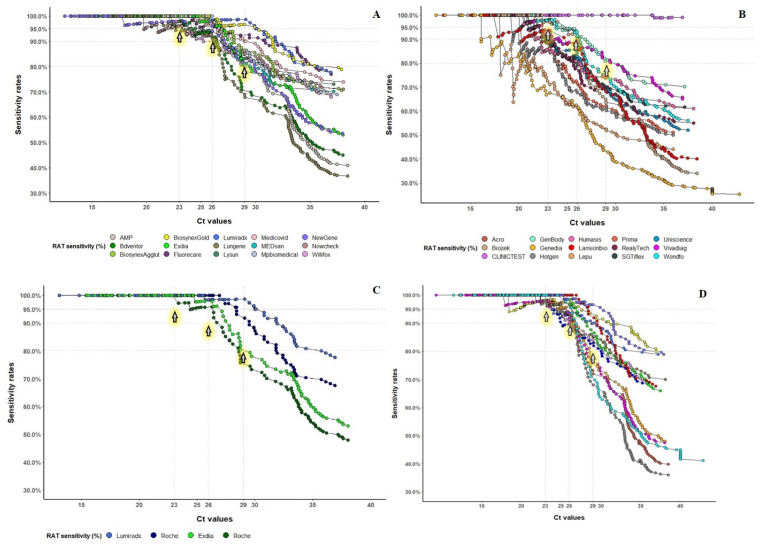
Comparative cumulated sensitivity curves for all rapid antigen tests (RATs) evaluated in the study. Figure 3 legend. (**A**). All RATs passing the validation criteria. (**B**). All RATs failing the validation. (**C**). RATs with an automated reader (Exdia from Precision Biosensor and LumiraDx assay from LumiraDx, Alloa) showed higher sensitivity performances compared to the internal reference (Roche). (**D**). Sensitivity curves comparing all the internal reference tests (Roche) used along each series showing the inter-laboratory and inter-series variability. RAT: rapid antigen test. On the *x*-axis there is the number of Ct; horizontal lines represent the Ct cut-offs considered for the validation criteria. Horizontal dotted lines and the three arrows represent the threshold of sensitivity rates to be considered for validation at different Ct cut-offs.

**Table 1 microorganisms-09-02589-t001:** Sensitivity and specificity rates of antigen tests passing the validation, stratified according to the viral load. Numbers in brackets indicate the 95% confidence intervals.

Manufacturer	Antigen Assay	Sensitivity at Ct ≤ 29	Sensitivity at Ct ≤ 26	Sensitivity at Ct ≤ 23	Specificity
		Ref. Value: ≥80% (CI) *	Ref. Value: ≥90% (CI)	Ref. Value: ≥95% (CI)	Ref. Value: ≥99%
BIOSYNEX Swiss	Biosnyex COVID-19 Ag + BSS °	86.5% (77.6, 92.8)	98.6% (92.4, 100)	100% (92.8, 100)	99.5%
BIOSYNEX Swiss	Biosynex COVID-19 Ag BSS ^	96.6% (90.5, 99.3)	98.6% (92.4, 100)	100% (92.8, 100)	100%
Newgene—Hangzhou-Bioengineering	COVID-19 Antigen Detection Kit	81.7% (69.6, 90.5)	94.1% (83.8, 98.8)	97.8% (88.2, 99.9)	99%
CLUNGENE	COVID-19 Rapid Antigen Test Cassette	69.6% (55.9, 81.2)	92.7% (80.1, 98.5)	94.3% (80.8, 99.3)	100%
Precision Biosensor, Inc.	Exdia COVID-19 Ag	80.6 (68.6, 89.6)	97.9% (88.7, 100)	100% (89.7, 100)	99.5%
Shenzhen Microprofit Biotech Co.	Fluorecare^®^ SARS-CoV-2 Spike Protein Test Kit	92.4% (84.2, 97.2)	95.8% (88.0, 99.1)	100% (93.0, 100)	100%
LumiraDx, Alloa	LumiraDx SARS-CoV-2 Ag	98.6% (92.6, 100)	98.3% (91.1, 100)	100% (91.2, 100)	99%
HANGZHOU LYSUN Biotechnology	LYSUN SARS-CoV-2 Antigen	83.2% (73.7, 90.3)	92.1% (83.6, 97.1)	97.9% (88.9, 100.0)	100%
Xiamen Boson Biotech Co., Ltd.	Medicovid-AG^®^ Test	90% (81.2, 95.6)	97.1% (89. 8, 99.6)	100% (92.8, 100)	99.3%
MEDsan GmbH	MEDsan^®^ SARS-CoV-2 Antigen Rapid Test	80.9% (71.2, 88.5)	92.1% (83.6, 97.1)	97.9% (88.9, 100)	100%
MP Biomedicals GmbH	MP COVID-19 Antigen Rapid Test	86.3% (76.3, 93.2)	100% (93.7, 100)	100% (92.1, 100)	100%
Bionote	NowCheck^®^ COVID-19 Ag Test	85% (75.3, 92.0)	92.7% (83. 7, 97.6)	95.9% (86.0, 99.5)	99%
AMP Diagnostics	Test rapide AMP SARS-CoV-2 Ag	79.3 (65.9, 89.2)	100% (90.97, 100)	100% (90.0, 100)	100%
Becton Dickinson	The BD Veritor™ System	70.9% (58.1, 81.8)	93.6% (82.5, 98.7)	100% (89.7, 100)	99.7%
Willifox	Willi Fox COVID-19 Antigen Test^®^	84.8% (75.0, 91.9)	90% (80.5, 95.9)	96.1% (86.5, 99.5)	100%

Legend: RAT: rapid antigen test. VL > 105: viral load corresponding to a cycle threshold (Ct) lower than 29. VL > 106: viral load corresponding to a cycle threshold (Ct) lower than 26. VL > 107: viral load corresponding to a cycle threshold (Ct) lower than 23. CI: confidence intervals. Ref.: FOPH reference. IR: internal reference for antigen test (Standard Q^®^, SD Biosensor/Roche). *: CI refers to the accuracy of the test, which, in this case (considering a population of all RT-PCR positive patients), corresponds to the sensitivity. °: agglutination test. ^: test with colloidal gold.

**Table 2 microorganisms-09-02589-t002:** Sensitivity and specificity rates of antigen tests failing the validation, stratified according to the viral load. Numbers in brackets indicate the 95% confidence intervals.

Manufacturer	Antigen Assay	Sensitivity at Ct ≤ 29	Sensitivity at Ct ≤ 26	Sensitivity at Ct ≤ 23	Specificity
		Ref. Value: ≥80% (CI) *	Ref. Value: ≥90% (CI)	Ref. Value: ≥95% (CI)	Ref. Value: ≥99%
Acro Biotech	Acro COVID-19 Antigen test	61.7% (50.3, 72.3)	72.5% (60.4, 82.5)	88.2% (76.1, 95.6)	100%
Healgen Scientific Limited Liability Company	CLINITEST^®^, Rapid COVID-19 Antigen Test ^+^	N/A	N/A	N/A	0%
PRIMA Lab SA	COVID-19 Antigen Rapid Test	61.7% (50.3, 72.3)	72.5% (60.4, 82.5)	88.2% (76.1, 95.6)	100%
BIOZEK	COVID-19 Antigen Rapid Test Cassette	66.0% (51.7, 78.5)	92.1% (78.6, 98.3)	96.9% (84.2, 99.9)	98.7%
GenBody Inc.	GenBody COVID-19 Ag	64.0% (53.2, 74.0)	72.4% (60.9, 82.0)	79.2% (65.0, 89.5)	100%
Green Cross Medical Science Corp.	Genedia W COVID-19 Ag ^§^	43.5% (30.9, 56.7)	62.8% (46.7, 77.0)	66.7% (49.8, 80.9)	100%
Humasis	Humasis COVID-19 Ag Test	73.8% (62.7, 83.0)	80.9% (69.5, 89.4)	85.7% (72.8, 94.1)	99.3%
Lansion Biotechnology Co.	Lansionbio^®^ COVID-19 Antigen Test Kit—Dry Fluorescence Immunoassay	69.4% (54.6, 81.8)	88.2% (72.6, 96.7)	93.3% (77.9, 99.2)	88.8%
Beijing Hotgen Biotech Co.	Novel Coronavirus 2019-nCoV Antigen Test—colloidal Gold	60.5% (49.0, 71.2)	68.7% (56.2, 79.4)	85.1% (71.7, 93.8)	100%
Hangzhou Realy Tech Co.	REALY antigen test	68.8% (57.4, 78.7)	77.9% (66.2, 87.1)	89.8% (77.8, 96.6)	100%
Lepu Medical Zechnology Co.	SARS-CoV-2 Antigen Rapid Test—Colloidal Gold	53.1% (41.7, 64.3)	62.3% (49.8, 73.7)	72.5% (58.3, 84.1)	100%
Wuhan UNscience Biotechnology Co.	SARS-CoV-2 Antigen Rapid Test Kit	69.9% (58.0, 80.1)	84.2% (72.1, 92.5)	88.9% (76.0, 96.3)	100%
Sugentech Inc.	SGTi-flex COVID-19 Ag	69.7% (59.0, 79.0)	80.3% (69.5, 88.5)	91.7% (80.0, 97.7)	100%
VivaCheck Biotech (Hangzhou) Co.	VivaDiag™, SARS-CoV-2 Rapid Ag Test	82.3% (72.1, 90.0)	88.6% (78.7, 94.9)	92.2% (81.1, 97.8)	100%
Guangzhou Wondfo Biotech Co.	Wondfo SARS-CoV-2 Antigen Test	75.3% (63.9, 84.7)	92.9% (83.0, 98.1)	97.8% (88.2, 99.9)	100%

Legend: RAT: rapid antigen test. VL > 105: viral load corresponding to a cycle threshold (Ct) lower than 29. VL > 106: viral load corresponding to a cycle threshold (Ct) lower than 26. VL > 107: viral load corresponding to a cycle threshold (Ct) lower than 23. CI: confidence intervals. Ref.: FOPH reference. IR: internal reference for antigen test (Standard Q^®^, SD Biosensor/Roche). *: CI refers to the accuracy of the test, which, in this case (considering a population of all RT-PCR positive patients), corresponds to the sensitivity. ^+^ The technical validation setup for this test did not allow to properly evaluate the performance of the antigen assay because it cross-reacted with physiological saline solution and generated a background band. ^§^ For this test, a sub-analysis of 41 samples was performed using only cobas6800© as RT-PCR reference, obtaining sensitivities of 50% (CI 29.9%–70%), 59.1% (CI 36.4%–79.3%) and 65% (CI 40.8%–84.6%) for VL above 105, 106 and 107, respectively).

**Table 3 microorganisms-09-02589-t003:** Sensitivity and specificity rates of Standard Q^®^, stratified according to the viral load, along all batches of validation. Numbers in brackets indicate the 95% confidence intervals.

Validation Batch	Sensitivity at Ct ≤ 29	Sensitivity at Ct ≤ 26	Sensitivity at Ct ≤ 23	Specificity
Serie 1	89.9% (93, 98.3)	93.4% (94.9, 99.3)	97.9% (97.2, 99.9)	100%
Serie 2	83.9% (74.1, 91.2)	91.3% (82, 96.7)	100% (93, 100)	100%
Serie 3	82.7% (72.7, 90.2)	89.5% (79.7, 95.7)	100% (92.5, 100)	100%
Serie 4	93.7% (85.8, 97.9)	97.1% (90.1, 99.7)	98% (89.6, 100)	100%
Serie 5	87.7% (77.9, 94.2)	100% (93.7, 100)	100% (92.1, 100)	100%
Serie 6	85% (75.3, 92.0)	94.1% (85.6, 98.4)	97.9% (89.2, 100)	99.5%
Serie 7	93.2% (84.7, 97.7)	100% (94.1, 100)	100% (91.2, 100)	99%
Serie 8	96.6% (90.5, 99.3)	98.6% (92.4, 100)	100% (92.7, 100)	99.0%
Serie 9	75.8% (63.3, 85.8)	95.7% (85.5, 99.5)	95.7% (85.5, 99.5)	99.7%
Serie 10	73.2% (59.7, 84.2)	95.1% (83.5, 99.4)	97.1% (85.1, 99.9)	100%
Serie 11	69.4% (54.6, 81.8)	94.1% (80.3, 99.3)	96.7% (82.8, 99.9)	100%
Serie 12	76.7% (64.0, 86.6)	90.2% (78.6, 96.7)	97.8% (88.2, 99.9)	99.5%
Serie 13	70.9% (58.1. 81.8)	97.7% (87.7, 99.9)	100% (90.9, 100)	100%

## Data Availability

Data supporting reported results will be available upon request for the peer-review process.

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
