# Peer review of "Multicenter Technical Validation of 30 Rapid Antigen Tests for the Detection of SARS-CoV-2 (VALIDATE)"

_microorganisms, 2021, doi:10.3390/microorganisms9122589_

Round 1

Reviewer 1 Report

Greub et al reported in this study that they developed a standardized technical validation protocol and assessed 30 RATs across four diagnostic laboratories and aimed at closing this gap by determining technical sensitivities and specificities of 30 RATs prior to market release for COVID test. They conclude that due to the high failure rate of 50%, a thorough validation of RATs prior to market release. Overall this study is well designed and presented. There are only a few minor comments.

  1. The author should clarify that how they distribute test samples to each RAT kits to minimize the sample difference.
  2. For Fig.2 legend, the author should label the meaning of the different color dot.
  3. The author should have the manuscript edited of English.

Author Response

Greub et al reported in this study that they developed a standardized technical validation protocol and assessed 30 RATs across four diagnostic laboratories and aimed at closing this gap by determining technical sensitivities and specificities of 30 RATs prior to market release for COVID test. They conclude that due to the high failure rate of 50%, a thorough validation of RATs prior to market release. Overall this study is well designed and presented. There are only a few minor comments.

The author should clarify that how they distribute test samples to each RAT kits to minimize the sample difference.

For Fig.2 legend, the author should label the meaning of the different color dot.

Reply: Thank you for your comment, we implemented the legend of Figure 2.

The author should have the manuscript edited of English.

Reply: Thank you for your comment, we had the manuscript edited for English.

Reviewer 2 Report

  1. The authors should note that the oversight in the USA is very different than the EU, as only a limited number of validated RATs are available, requiring FDA EUA prior to marketing.
  2. Some detail as to how samples were processed/extracted and whether the tests were LFTs or FIAs or something else needs to be included.
  3. Did the authors not test the Abbott and Quidel RATs that are heavily used in the US and elsewhere?

Author Response

The authors should note that the oversight in the USA is very different than the EU, as only a limited number of validated RATs are available, requiring FDA EUA prior to marketing.

Some detail as to how samples were processed/extracted and whether the tests were LFTs or FIAs or something else needs to be included.

Reply: All assays were detailed in Supplementary Table 1, including the type of test (LFT/FIA).

Did the authors not test the Abbott and Quidel RATs that are heavily used in the US and elsewhere?

Reply: Thank you for your comment. We did test the Abbott assay, indeed , but the results were not shown because it was not part of the validation protocol and it has been previously validated, together with the Standard Q from Roche, in previous studies (doi:10.1101/2020.11.20.20235341, doi:10.1101/2020.11.23.20237057). We did not test the assay from Quidel.

For clarity, we added the following paragraph:

“The Standard Q® (as well as the PanbioTM COVID-19 Ag Rapid Test from Abbott, USA) was validated in previous clinical studies [10, 18] and already commercialized at the time of the beginning of the present study. In addition, results from a technical validation of the PanbioTM assay were presented elsewhere [4]. The Standard Q® was chosen as internal reference based on its more frequent use. “

Reviewer 3 Report

Points of criticism/questions:

  1. The Standard Q® (SD Biosensor/Roche) assay is mostly referred to in the manuscript as “internal reference” (or as “internal reference standard” (p.3, l.107)). It should be consistently referred to as “reference standard” based on the SGM-SSM “Standard operating procedure to validate SARS-CoV-2 antigen tests”. The term “internal” is misleading.
  2. It would be helpful if Table 1 had a ranking based on sensitivity/specificity (e.g. based on sensitivity at Ct ≤29).
  3. Materials and Methods, p.2. ll.88/89: “The only exceptions were reference samples shared between all centers (Table S2). What does ”exceptions” refer to?
  4. Materials and Methods, p.2. ll.89/90: “Five highly positive reference samples were included.” A more precise wording would be helpful: “ The SARS-CoV-2 RT-PCR positive samples included five highly positive reference samples” or “Five of the SARS-CoV-2 RT-PCR positive samples were highly positive.”
  5. Materials and Methods, p.2. ll.98: “All reference samples were evaluated with…” Were both positive and negative or only the negative reference samples evaluated with the Biofire® FilmArray® Respiratory panel (bioMérieux), a SARS-CoV-2 specific quantitative RT-PCR and a virus specific quantitative RT-PCR?
  6. Sections 2.4 and 2.5 of the Materials and Methods section could be subsections of section 2.3.
  7. Materials and Methods 2.6: “transport media volumes slightly differed” (p.3, l.131). Mention this point in the discussion (“differences between centers”).
  8. Results, p.5, l.184: “3.2 Sensitivity of RATs. Fifteen out of 30 RATs passed the validation (Table 1).” Move the sentence to the next line.
  9. Results, p.5, ll.185/186: “Table 1. Sensitivity and specificity rates of antigen tests passing the validation, stratified according to the viral load. Numbers in brackets indicate the 95% confidence intervals” Move the second sentence to the legend of Table 1 (ditto Table 2).
  10. Results, p.5, ll.196/197: “Twelve assays passed the FOPH validation criteria.” This is in contradiction to the previous sentence. (p.5, l.184: Fifteen out of 30 RATs passed the validation (Table 1).”
  11. Results, p.7, ll.218/219: “Three RATs (AMP Diagnostics, LUNGENE and Becton Dickinson) were validated based on non-inferiority criteria (sensitivity ≤ 5%) to the internal reference.” In the Materials and Methods section 2.7, > 3 assays are listed as assays validated based on non-inferiority criteria compared to the reference test.
  12. Discussion: On the one hand you state that “we did not access patient charts during this technical validation” (p.9, ll.294/205), but on the other you mention that “not every patient included in the study had a COVID-19 symptoms duration shorter than four days.” (p.9, ll.295/296). Thus, it is not clear if, at least retrospectively, you had access to the patient data. If yes, patient characteristics, in particular with regard to symptoms (present? duration?), should be given in a table.
  13. The “top performing” RATs (p8, l.234) should be mentioned in the discussion.

Author Response

The Standard Q® (SD Biosensor/Roche) assay is mostly referred to in the manuscript as “internal reference” (or as “internal reference standard” (p.3, l.107)). It should be consistently referred to as “reference standard” based on the SGM-SSM “Standard operating procedure to validate SARS-CoV-2 antigen tests”. The term “internal” is misleading.

Authors' reply: Thank you for this suggestion. We modified the manuscript accordingly.

It would be helpful if Table 1 had a ranking based on sensitivity/specificity (e.g. based on sensitivity at Ct ≤29).

Authors' reply: Thank you for your useful comment. We classified the assays in a descending order according to the sensitivity scores at Ct ≤29.

Materials and Methods, p.2. ll.88/89: “The only exceptions were reference samples shared between all centers (Table S2). What does ”exceptions” refer to?

Authors' reply: The exceptions were the reference samples shared between centers, which were stored longer and not necessarily used within the 72hours from the collection. We clarified this in the text.

Materials and Methods, p.2. ll.89/90: “Five highly positive reference samples were included.” A more precise wording would be helpful: “ The SARS-CoV-2 RT-PCR positive samples included five highly positive reference samples” or “Five of the SARS-CoV-2 RT-PCR positive samples were highly positive.”

Authors' reply: The sentence was rephrased accordingly.

Materials and Methods, p.2. ll.98: “All reference samples were evaluated with…” Were both positive and negative or only the negative reference samples evaluated with the Biofire® FilmArray® Respiratory panel (bioMérieux), a SARS-CoV-2 specific quantitative RT-PCR and a virus specific quantitative RT-PCR?

Authors' reply: Thank you for your specification. Indeed, we referred to negative samples. We clarified it in the text.

Sections 2.4 and 2.5 of the Materials and Methods section could be subsections of section 2.3.

Authors' reply: The sub-sections were modified accordingly.

Materials and Methods 2.6: “transport media volumes slightly differed” (p.3, l.131). Mention this point in the discussion (“differences between centers”).

Authors' reply: Thank you for your comments, the following sentence was added: “Furthermore, as previously mentioned, transport media volumes slightly differed be-tween centers and this could have played a limited role in RATs performances.”

Results, p.5, l.184: “3.2 Sensitivity of RATs. Fifteen out of 30 RATs passed the validation (Table 1).” Move the sentence to the next line.

Authors' reply: Modified accordingly.

Results, p.5, ll.185/186: “Table 1. Sensitivity and specificity rates of antigen tests passing the validation, stratified according to the viral load. Numbers in brackets indicate the 95% confidence intervals” Move the second sentence to the legend of Table 1 (ditto Table 2).

Authors' reply: Modified accordingly.

Results, p.5, ll.196/197: “Twelve assays passed the FOPH validation criteria.” This is in contradiction to the previous sentence. (p.5, l.184: “Fifteen out of 30 RATs passed the validation (Table 1).”

Authors' reply: 15/30 RATs passed the validation: Twelve passing the FOPH validation criteria and three RATs (AMP Diagnostics, LUNGENE and Becton Dickinson) were validated based on non-inferiority criteria (sensitivity ≤ 5%) to the reference standard.

Results, p.7, ll.218/219: “Three RATs (AMP Diagnostics, LUNGENE and Becton Dickinson) were validated based on non-inferiority criteria (sensitivity ≤ 5%) to the internal reference.” In the Materials and Methods section 2.7, > 3 assays are listed as assays validated based on non-inferiority criteria compared to the reference test.

Authors' reply: Yes, but only three passed the validation. We reformulated the sentence for clarity purposes as follows: “three RATs (AMP Diagnostics, LUNGENE and Becton Dickinson) passed the validation based on non-inferiority criteria (sensi-tivity ≤ 5%) to the reference standard”:

Discussion: On the one hand you state that “we did not access patient charts during this technical validation” (p.9, ll.294/205), but on the other you mention that “not every patient included in the study had a COVID-19 symptoms duration shorter than four days.” (p.9, ll.295/296). Thus, it is not clear if, at least retrospectively, you had access to the patient data. If yes, patient characteristics, in particular with regard to symptoms (present? duration?), should be given in a table.

Authors' reply: Thank you for your comment. We could not provide a table because, as stated, we did not access to patients’ charts during this study. The only clinical criteria considered was the presence of COVID-19 symptoms (reported directly in the patients’ SARS-CoV2 analysis request). Some of the patients (from one out of four centers) included in the present study had a longer duration of symptoms: this information was obtained as part of another study where we could collect more specific clinical details, but these results are not yet published.

Authors' reply: In the samples section we added the specification about the symptomatic status of patients included in the study.

The “top performing” RATs (p8, l.234) should be mentioned in the discussion.

Authors' reply: The discussion was implemented accordingly: “Among these, the “top performer” assay was the LumiraDx SARS-CoV-2 Ag test (Lu-miraDx, Alloa).”

Reviewer 4 Report

In their paper, Greub et al use a standardized validation protocol to test 30 different rapid antigen tests for the detection of SARS-CoV2. They show that only half of the RATs actually obtain the required sensitivity as set by the Swiss FOPH. The paper is well written and the data is easy to interpret. I have two comments that could improve the paper:

1) Rapid antigen tests are not quantitative and a call of positive or negative must be made. The main text should address how assessment was made for different RATs and how this may affect the interpretation. 

2) A discussion point that RATs with high specificity but lower sensitivity (~60%) are still preferable to no test at all, they are just non-preferable to better RATs or RT-PCR assays.

Author Response

In their paper, Greub et al use a standardized validation protocol to test 30 different rapid antigen tests for the detection of SARS-CoV2. They show that only half of the RATs actually obtain the required sensitivity as set by the Swiss FOPH. The paper is well written and the data is easy to interpret. I have two comments that could improve the paper:

1) Rapid antigen tests are not quantitative and a call of positive or negative must be made. The main text should address how assessment was made for different RATs and how this may affect the interpretation.

Reply: Thank you for your comment. The different types of reading were disclosed in lines 124-125. The names of assays with automated readers were added in lines 230-231. A comment about the limits of visual reading was added: “Finally, it needs to be taken into account that visual reading, compared to automated one, could sometimes result subjective, according to the reader’s experience.”

2) A discussion point that RATs with high specificity but lower sensitivity (~60%) are still preferable to no test at all, they are just non-preferable to better RATs or RT-PCR assays.

Reply: Thank you for your comment. We think that this kind of statement might be misleading as well as misinterpreted and for this reason we preferred to limit the discussion to the technical limits of the protocol and tests' results.